# The Potential Benefits of Curcumin-Enriched Diets for Adults with Colorectal Cancer: A Systematic Review

**DOI:** 10.3390/antiox14040388

**Published:** 2025-03-26

**Authors:** María Neira, Constanza Mena, Keila Torres, Layla Simón

**Affiliations:** Escuela de Nutrición y Dietética, Universidad Finis Terrae, Santiago 7501015, Chile; mneiram@uft.edu (M.N.); cmenaf@uft.edu (C.M.)

**Keywords:** *Curcuma*, colorectal adenoma, anticancer diets, anti-inflammatory agents, antioxidants, gastrointestinal health

## Abstract

Colorectal cancer (CRC) is the second leading cause of cancer-related deaths worldwide. Conventional treatments such as chemotherapy and radiotherapy are often associated with severe side effects and limited effectiveness. Curcumin, a polyphenol derived from *Curcuma longa*, has demonstrated anti-inflammatory and anticancer properties. A systematic review of the recent scientific literature followed PRISMA guidelines to evaluate the benefits of a curcumin-enriched diet for adults with colorectal cancer. Articles published between 2018 and 2024 were retrieved from PubMed, SciELO, Google Scholar, and Scopus. Studies meeting the inclusion criteria focused on curcumin, adults, and colorectal cancer outcomes. The administration of curcumin-containing products was associated with improved survival rates, enhanced quality of life, tumor reduction, and anti-inflammatory effects. A curcumin-enriched diet shows potential as an effective adjunct therapy for CRC patients, though its limited bioavailability and potential side effects, such as gastrointestinal discomfort, pose challenges. Addressing these limitations through larger cohorts, extended study durations, and improved formulations to enhance bioavailability is essential. Such efforts could enable the development of personalized dietary recommendations for CRC management.

## 1. Introduction

Colorectal cancer (CRC) is one of the most prevalent malignancies globally, standing as the third most frequently diagnosed cancer and the second leading cause of cancer-related deaths worldwide [1]. In 2020, approximately 1.9 million new cases and 930,000 deaths were reported. Projections for 2040 indicate a significant increase, with an estimated 3.2 million cases and 1.6 million deaths [2]. Notably, the proportion of CRC cases diagnosed and deaths occurring among individuals under 50 years has risen significantly in recent decades [3,4].

CRC develops from precancerous polyps, such as tubular adenomas or serrated polyps, driven by disruptions in DNA repair and cell proliferation [5]. The disease progresses from early stages with localized tumors (Stage I) to advanced stages with metastasis (Stage IV) [6]. Risk factors for CRC encompass a combination of inherited and acquired causes. Genetic predisposition, including a family history of colorectal polyps or cancer, markedly increases the risk of developing CRC. Approximately 10–20% of CRC cases are associated with a positive family history, while 5% are linked to hereditary CRC syndromes identifiable through germline testing [7]. Abnormalities in the colonic epithelium and chronic inflammatory conditions, such as inflammatory bowel disease, ulcerative colitis, or Crohn’s disease, are also important contributors to CRC development [8]. Additionally, dietary exposure, particularly from diets low in fiber and high in fat, along with the consumption of red and processed meats, significantly increases the risk. Lifestyle factors, such as obesity, tobacco use, and alcohol consumption, further exacerbate this risk [9].

Current diagnostic interventions for CRC include fecal blood tests, blood analysis, colonoscopy, proctoscopy, ultrasounds, and biopsies [10]. Early-stage treatment often involves the surgical removal of polyps or endoscopic procedures, while advanced cases may require partial colectomy [11]. Non-surgical treatments, such as immunotherapy, chemotherapy, and radiotherapy, play vital roles in CRC. Immunotherapy targets cancer cells through immune system enhancement, particularly effective in MSI-H (microsatellite instability-high) or dMMR (deficient mismatch repair) tumor phenotypes. Chemotherapy encompasses the use of cytotoxic drugs to reduce tumor size and delay the progression, while radiotherapy applies high-energy radiation to shrink or eliminate tumors pre- or post-surgery [12]. However, these conventional therapies are frequently associated with significant adverse effects and a high risk of disease recurrence [13]. In advanced cases, when other options are not viable, palliative care is provided to improve quality of life, including nutritional interventions focused on understanding how body weight, diet, and physical activity influence cancer risk and progression [14,15]. Among these, curcumin has attracted significant attention for its potential role in the prevention and treatment of CRC [16,17,18,19].

Curcumin, a natural compound extracted from the rhizome of the Asian turmeric plant *Curcuma longa* or *xanthorrhiza* (both belonging to the ginger family (Zingiberaceae)), is widely recognized for its distinctive yellow color. This compound has long been utilized in dietary supplements, food additives, and various products, including dairy, cereals, mustard, pickles, sausages, meat, fish, eggs, bakery items, and beverages [20,21]. Curcumin is highly sensitive to degradation in a wide range of pH (between pH 3 and 10), and its decomposition occurs faster under neutral and alkaline conditions. It exhibits minimal solubility at room temperature in aqueous solutions at a neutral or acidic pH. Nevertheless, its lipophilic properties enable its dissolution in a variety of organic solvents, including methanol, ethanol, acetone, and dimethyl sulfoxide (DMSO) [22]. Curcumin is known for its antioxidant, anti-inflammatory, anticoagulant, antimicrobial, and anticancer properties [23,24,25]. Curcumin supplementation enhances antioxidant activity in adults by increasing total antioxidant capacity, decreasing malondialdehyde levels, and modulating superoxide dismutase activity [26]. In normal cells, curcumin prevents carcinogenesis through its antioxidant activity by quenching free radicals and inducing antioxidant enzymes (superoxide dismutase, catalase, and glutathione peroxidase) and protein markers (Nrf2 and HO-1). On the other hand, curcumin acts as a pro-oxidant in cancer cells by inducing reactive oxygen species (ROS) accumulation, promoting DNA damage and apoptosis while enhancing chemotherapeutic efficacy by sensitizing drug-resistant cells. In this way, curcumin has a dual role in cancer as an antioxidant agent participating in prevention and as a pro-oxidant effector with beneficial effects in treatment [27].

Curcumin has demonstrated the potential to inhibit tumor progression across various cancers, enhance the quality of life, and behave as a complementary therapy to conventional treatments [28,29,30]. Specifically in CRC, Ojo et al. (2022) reviewed the literature on the mechanisms of the action of curcumin, highlighting its ability to induce apoptosis, cell cycle arrest, and autophagy while inhibiting inflammation and angiogenesis. Among their conclusions, the authors emphasized the need for larger clinical trials to understand better the dosage, toxicity, bioavailability, and therapeutical application of curcumin in CRC [31]. Despite these benefits, the clinical use of curcumin remains limited due to its poor solubility and low bioavailability [32]. A systematic, transparent, and reproducible review of the literature is necessary to assess the clinical benefits of curcumin-enriched diets in adult patients with CRC. Accordingly, this systematic review aims to consolidate current evidence on the therapeutic benefits of a curcumin-enriched diet for CRC patients.

## 2. Methods

### 2.1. Study Design

Our systematic review was designed according to the Preferred Reporting Items for Systematic Reviews and Meta-Analyses (PRISMA) guidelines. This study was registered in the Open Science Framework (Curcumin-Enriched Diets for Adults with Colorectal Cancer: A Systematic Review registered on https://doi.org/10.17605/OSF.IO/JXKD9 on 23 December 2024). Since it was based on analyzing previously published data, it did not require ethical approval.

### 2.2. Search Strategy

The systematic review was conducted between March and May 2024 using PubMed, SciELO, Google Scholar, and Scopus as search databases. Two researchers independently extracted the data to ensure accuracy and reliability. The research question, refined after assessing its scope and feasibility, was as follows: What are the beneficial effects of a curcumin-enriched diet in adult patients with colorectal cancer? This question was structured using the PICOS framework, where P (Population) corresponds to adults with colorectal cancer, I (Intervention) refers to a curcumin-enriched diet, C (Comparison) represents a diet without curcumin, O (Outcome) focuses on the beneficial effects for colorectal cancer, and S (Setting) specifies the context of curcumin and colorectal cancer.

Keywords and search terms were defined in both English and Spanish to maximize the scope of the search. These included “Curcumin/Curcumina”, “Adults/Adultos”, “Colorectal cancer/Cáncer colorrectal”, “Benefits/Beneficios”, and “Gastrointestinal system/Sistema gastrointestinal”. Search queries combined these terms using Boolean operators, resulting in equations such as curcumin AND Adult AND Benefits AND (“colorectal cancer” OR “gastrointestinal”) and curcumina AND Adultos AND beneficios AND (“cancer colorrectal” OR “gastrointestinal”) for searches in the databases.

### 2.3. Selection Criteria

The inclusion criteria used in this study were articles published between 2018 and 2024 in English or Spanish (the languages of the authors); focus on colorectal cancer patients and models; and interventions involving curcumin-containing products. On the other hand, the exclusion criteria for selecting research articles were studies involving anticoagulant users, children, adolescents, or pregnant women; and studies that did not report original data.

### 2.4. Data Analysis

Data on survival rates, quality of life, tumor reduction, and inflammatory markers were extracted and analyzed. Excel software was employed to organize and manage the selection process, including identifying duplicate entries and screening articles based on their titles and abstracts.

## 3. Results

A total of 129 articles were identified during the initial search process. In the first screening phase, 19 duplicate articles were removed. Subsequently, in the second screening phase, an evaluation of titles and abstracts excluded an additional 95 articles. Ultimately, 15 original articles met the inclusion criteria and were used for the final analysis (Figure 1).

The characteristics of the selected articles are summarized in Table 1. Among all the studies included in this review, eight were clinical trials, two were observational studies, three were in vitro studies (one also included an in vivo component), one was an ex vivo explant model study, and one was an in silico study. Of the 15 studies, 10 reported patient data. Among these, eight observed the positive beneficial effects of a curcumin-enriched diet in adult patients, while two reported unclear or negative results.

## 4. Discussion

Understanding the potential benefits of curcumin-containing products in the management of CRC in adults is essential for evaluating its feasibility as a non-invasive and supportive treatment option. This systematic review focuses on assessing the potential advantages of incorporating a curcumin-enriched diet in adult patients with CRC.

### 4.1. The Positive Impact of Curcumin-Enriched Diets on CRC Patients

Eight studies demonstrated that curcumin-containing products exert significant beneficial effects on colorectal and gastrointestinal cancers, particularly improving survival, reducing tumor size through apoptosis and oxidative stress in tumor cells, mitigating inflammation, enhancing the quality of life, and supporting conventional treatments [35,36,37,41,42,43,44,45]. These findings underscore the therapeutic potential of curcumin-enriched diets in CRC (Figure 2).

#### 4.1.1. Survival Improvement

Curcumin bioavailability is a critical factor in evaluating the impact of curcumin on patient survival. One study found no association between plasma curcumin levels and tumor-free intervals due to low plasma concentrations (0.003–0.050 µM), which were significantly below the effective range observed in preclinical models (5–50 µM) [38]. Conversely, two studies reported improved long-term survival when combining curcumin-containing products with chemotherapy. The combination of Bevacizumab/FOLFIRI (folinic acid, fluorouracil, and irinotecan) with ginsenoside-modified nanostructured lipid carriers (G-NLC) increased curcumin bioavailability, enhancing its solubility, stability, and cytotoxicity against CRC cells [41]. Similarly, a curcumin-containing product combined with Bevacizumab/FOLFOX (folinic acid, fluorouracil, and oxaliplatin) was safe, tolerable, and effective in improving overall survival in patients with metastatic CRC [35]. In addition, dietary intake studies suggest that a curcumin-enriched diet, as part of a mixed spice diet, provides protective effects against gastrointestinal cancers, particularly at low-to-medium intake levels [39]. However, limitations such as small cohort sizes, variations in curcumin-containing product formulations, and inconsistent dosing regimens may contribute to survival biases.

#### 4.1.2. Tumor Reduction

Khan et al. (2022) demonstrated that curcumin treatment induced apoptosis in both tumor and stromal tissues while enhancing T-cell movement toward the tumor margin, indicating a less immunosuppressive tumor microenvironment and a shift toward improved immune responses [45]. Similarly, Ahmed et al. (2020) reported that curcumin effectively reduced the tumor size in a xenograft mouse model by inducing apoptosis and inhibiting epithelial–mesenchymal transition (EMT), a key process in cancer progression and metastasis. Specifically, curcumin promotes cell death by increasing ROS production and suppressing the activation of survival signaling pathways [44]. Maletzki et al. (2019) investigated the combined effects of curcumin and low-dose chemotherapy in CRC models, demonstrating that curcumin exerts potent cytotoxic effects by inducing apoptosis and inhibiting tumor cell proliferation through oxidative stress-related mechanisms [43]. Another study found that curcumin suppressed tumor cell proliferation and enhanced apoptosis in CRC cells [42]. However, in a randomized trial with patients suffering from familial adenomatous polyposis, curcumin supplementation did not significantly reduce the number or size of adenomas, indicating limited efficacy in some tumor-related conditions [33]. Greil et al. (2018) did not observe significant anti-tumor activity in a Phase I trial using liposomal curcumin; however, tumor marker responses and clinical benefits were noted in some patients [34]. In conclusion, pure curcumin was effective in reducing tumor growth and modulating tumor-related pathways in CRC [45]. However, when curcumin was administrated in a liposomal formulation, no significant changes in tumor size were observed [34].

#### 4.1.3. Anti-Inflammatory and Antioxidant Effects

Patients with CRC often exhibit chronic inflammation during both the initiation and progression of the tumor. Scientific evidence suggests that curcumin-containing products, whether used alone or in combination with other therapies, exert anti-inflammatory effects in patients with cancer. Panahi et al. (2021) demonstrated that curcumin supplementation significantly reduced the serum levels of inflammatory markers, such as C-reactive protein and erythrocyte sedimentation rate, in stage 3 CRC patients [36]. Furthermore, a curcumin-containing product combined with anthocyanins modulated inflammatory biomarkers associated with colon carcinogenesis, although no direct changes were observed in the circulating inflammatory markers [40]. Also, a curcumin-containing product reduced NF-κB expression, a key regulator of inflammation, in adenoma tissues, suggesting a reduction in inflammation [37]. In an in vitro study, Maletzki et al. (2019) reported that curcumin exerted significant anti-inflammatory effects by inducing apoptosis and reducing immunosuppressive pathways in CRC cells, particularly when combined with Gemcitabine [43]. Nadeem et al. (2023) analyzed dietary factors influencing gastrointestinal cancers and found that dietary curcumin intake, along with other antioxidant-rich foods, exhibited protective effects against CRC development [39].

#### 4.1.4. Support to Conventional Treatment

Curcumin has shown the potential to enhance the efficacy of conventional cancer treatments in colorectal cancer. Adding a curcumin-containing product (2 g/day) to FOLFOX chemotherapy in patients with metastatic colorectal cancer significantly improved overall survival (OS) compared to chemotherapy alone. The combination was safe and well-tolerated. The hazard ratio (HR) for OS was 0.34 (95% CI: 0.14, 0.82; *p* = 0.02), with median survival times of 200 days for FOLFOX alone and 502 days for the combination of the curcumin-containing product and FOLFOX. However, the progression-free survival (PFS) improvement was not statistically significant [35]. Another study that combined a curcumin-containing product with Bevacizumab/FOLFIRI reported significant improvements in survival outcomes and tumor responses. The median OS was 30.7 months, and the median PFS was 12.8 months for the combination of curcumin with Bevacizumab/FOLFIRI [41]. Combining curcumin with Gemcitabine significantly enhanced tumor elimination in colorectal cancer cells, showcasing its synergistic potential with chemotherapy agents [43]. However, research evaluating the addition of a curcumin-containing product to pre-surgery chemoradiation therapy in locally advanced rectal cancer patients found no improvement in pathologic complete response rates, likely due to variable bioavailability [38]. These findings suggest that curcumin can complement conventional CRC treatments by enhancing survival outcomes and tumor responses, though challenges related to bioavailability require further investigation.

#### 4.1.5. Quality of Life

To evaluate the quality of life, neurotoxicity, and inflammation in patients, a study was conducted involving the administration of 500 mg curcumin-containing capsules daily for 8 weeks. A questionnaire assessed five functional domains (physical, cognitive, emotional, social, and role abilities), nine symptom domains, and overall quality of life. This tool was administered at the beginning and after 8 weeks of treatment. The results demonstrated significant improvements in global quality of life and functional scores in the curcumin-treated group compared to the control group [36]. Although no significant differences in quality of life or neurotoxicity were observed between groups receiving the curcumin-containing product and those receiving FOLFOX alone, the addition of curcumin was safe and tolerable, indicating its feasibility as a complementary treatment [35]. The use of curcumin as a complementary medicine has also been linked to psychological benefits, including improved resilience and spirituality, which contribute to a healthier lifestyle for cancer patients [15]. In summary, two studies demonstrate the positive effects of curcumin-containing products on quality of life and psychological status; meanwhile, another study was unable to detect significant effects. These findings suggest that curcumin supplementation may alleviate some of the physical and functional burdens of CRC, enhancing the overall well-being of patients.

### 4.2. Mechanisms of Action of Curcumin-Enriched Diets in CRC

Curcumin-enriched diets exhibit multiple mechanisms of action in CRC, demonstrating significant anticancer potential. Curcumin modulates signaling pathways and molecular targets involved in tumor growth and progression. Curcumin upregulates miR-491, which inhibits the PEG10 and Wnt/β-catenin signaling pathways, thereby reducing tumor proliferation and promoting apoptosis in CRC cells [42]. Similarly, curcumin-induced oxidative stress triggers apoptosis while inhibiting EMT, a key process in cancer metastasis, by targeting the PRP4 kinase domain [44]. Additionally, the expression of SYK has been implicated in various patterns of CRC by initiating tumor metastasis but could be targeted by curcumin [46] (Figure 3).

In terms of immune modulation, curcumin has been observed to reduce immunosuppressive tumor microenvironments by inducing apoptosis in tumor and stromal tissues and enhancing T-cell migration toward the tumor margin [45]. Furthermore, curcumin suppresses NF-κB expression. Although a trend toward reduced Ki-67 expression, an established tumor proliferation marker, was observed, these changes did not reach statistical significance [37]. These findings illustrate the ability of curcumin to target molecular pathways, modulate immune responses, and suppress inflammatory signals.

### 4.3. Adverse Events of a Curcumin Diet in Adult Patients with CRC

While curcumin-enriched diets have demonstrated potential anticancer effects in CRC, several studies have reported side effects and challenges associated with curcumin administration (Figure 4). Oral curcumin supplementation has been associated with gastrointestinal discomfort in some patients. One study reported low rates of adverse events (AEs) over a 4- to 6-week period, including dyspepsia (2/10; 20%), stomach pain (1/10; 10%), nausea (1/10; 10%), and diarrhea (1/10; 10%). However, it could not be definitively established that these AEs were directly caused by the combined treatment with the curcumin-containing product and anthocyanins, as similar effects were also observed in the placebo group [37].

A study involving 18 patients receiving FOLFOX chemotherapy combined with an oral curcumin-containing product (CUFOX) reported various gastrointestinal AEs potentially attributable to curcumin. These included diarrhea (10/18 patients), nausea (8/18), dyspepsia (7/18), oral mucositis (6/18), vomiting and constipation (5/18 each), and anorexia (4/18). Less common symptoms included abdominal pain, acute kidney injury, flatulence, bloating, and dry mouth [35]. In another study, administering 500 mg/day of a curcumin-containing product for 8 weeks in patients undergoing chemotherapy for CRC did not exacerbate gastrointestinal symptoms beyond what was expected from their prior treatments. Moreover, curcumin supplementation was reported to improve the quality of life in these patients [36].

Cruz-Correa et al. (2018) reported that one patient experienced allergic reactions (pruritus), and the participant was promptly withdrawn from the protocol [33]. Hepatotoxicity and hematological effects have been reported in rare instances with liposomal curcumin administration, particularly at high doses and in patients with pre-existing conditions [34]. Jeon et al. (2022) reported that the most common Grade 3 or higher AEs were neutropenia (34.1%, n = 15), followed by nausea and vomiting (both 9.1%, n = 4). In contrast, anemia, diarrhea, and stomatitis were observed at Grade 2 or lower [41].

A curcumin-containing product has synergistically enhanced the efficacy of chemotherapies like FOLFOX. However, it may occasionally worsen chemotherapy-induced side effects and potentially interact with the drug [35]. In conclusion, while curcumin supplementation has been associated with gastrointestinal discomfort [35], allergic reactions [33], and anemia [41] in some patients, these findings remain inconclusive [37]. Variability in reported side effects may arise from differences in dosage, formulation, conventional chemotherapy-associated AEs, and individual patient characteristics, underscoring the need for further research to establish clearer safety and efficacy profiles.

### 4.4. Limitations and Challenges

The main limitation identified in this review was the small sample size and short duration of several studies. Two studies had fewer than 40 participants and durations of less than 8 weeks [35,36], limiting the ability to assess long-term outcomes effectively. Additionally, the poor water solubility of curcumin leads to low bioavailability, complicating the assessment of whether adverse effects stem from curcumin administration itself or the prior treatments received by patients. This challenge has been noted in studies investigating both oral and intravenous formulations [34,38,41]. Furthermore, a study on familial adenomatous polyposis found no significant clinical benefits, suggesting that bioavailability challenges may limit the therapeutic potential of curcumin in certain contexts [33].

Future research should prioritize studies with sample sizes of 40 or more participants and durations of at least 12 months to enable detailed investigations into the long-term side effects of curcumin administration in adult patients with CRC. Additionally, research should explore the clinical, pathological, and molecular heterogeneity of CRC and the mechanisms underlying the curcumin action. This comprehensive approach would provide a clearer understanding of the curcumin benefits and support the development of personalized recommendations tailored to each age group.

### 4.5. Current State of Knowledge

Recently, Gutsche et al. (2025) conducted a systematic review and meta-analyzed the clinical evidence of potential benefits and risks associated with curcumin supplementation in head and neck, breast, prostate, and colorectal cancer. However, the included studies showed considerable heterogeneity, making it difficult to undertake a direct comparison and conclude about the effectiveness of curcumin on cancer treatment [47]. Yuan et al. (2025) developed 5-fluorouracil–curcumin loaded in a silk fibroin hydrogel, which exhibited low cytotoxicity toward normal human colon epithelial cells while significantly increasing the apoptotic rate and reducing the viability in CRC cells [48]. Moreover, curcumin and metformin have been reported to induce lipid peroxidation and ferroptosis in CRC cells, leading to ROS accumulation and oxidative stress, which activate autophagy and ferroptosis, ultimately causing cell death [49]. According to the evidence, curcumin may have anti-tumoral effects in CRC that are poorly extrapolatable to other types of cancers. Additionally, the poor bioavailability of curcumin suggests that the positive effects described in our systematic review may derive from a local effect in the colon. Overall, the current state of knowledge highlights oral curcumin-containing product administration as a potential non-invasive therapeutic option for the treatment of CRC.

## 5. Conclusions

This systematic review highlights the potential benefits of a curcumin-enriched diet in adult patients with CRC, focusing on its therapeutic effects, mechanisms of action, and possible long-term side effects. The findings indicate that curcumin-containing products may improve survival rates, enhance quality of life, and reduce tumor size in patients. Its mechanisms of action include inhibiting cellular proliferation and inducing apoptosis and oxidative stress. However, adverse effects, such as gastrointestinal disturbances, anemia, and pruritus, were also observed. Despite these promising outcomes, the evidence remains limited due to challenges such as low bioavailability, small sample sizes, and the short duration of many studies. These limitations emphasize the need for more rigorous investigations to fully evaluate curcumin’s therapeutic potential.

A curcumin-enriched diet could play a supportive role alongside conventional treatments, such as chemotherapy and radiotherapy, in CRC management. However, the current scientific evidence is insufficient to establish its use as a dietary intervention. Future studies should focus on larger, more diverse populations, extended intervention periods, and improved methods to enhance curcumin bioavailability to provide stronger and more complete conclusions.

## Figures and Tables

**Figure 1 antioxidants-14-00388-f001:**
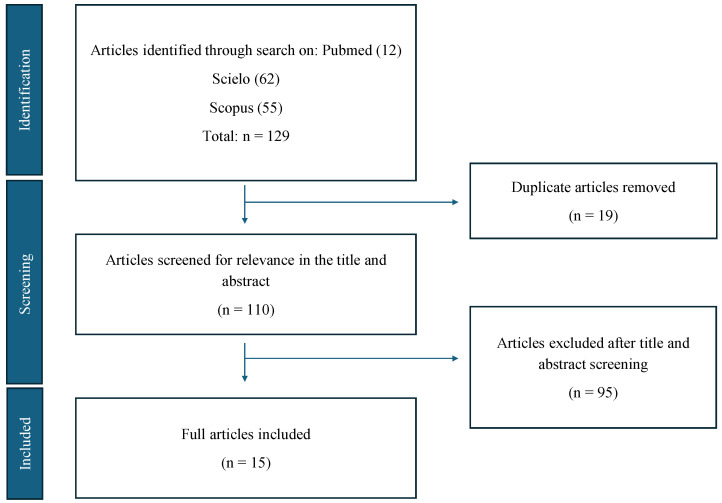
Flow diagram illustrating the identification and selection process of articles for the study, following the Preferred Reporting Items for Systematic Reviews and Meta-Analyses (PRISMA) guidelines.

**Figure 2 antioxidants-14-00388-f002:**
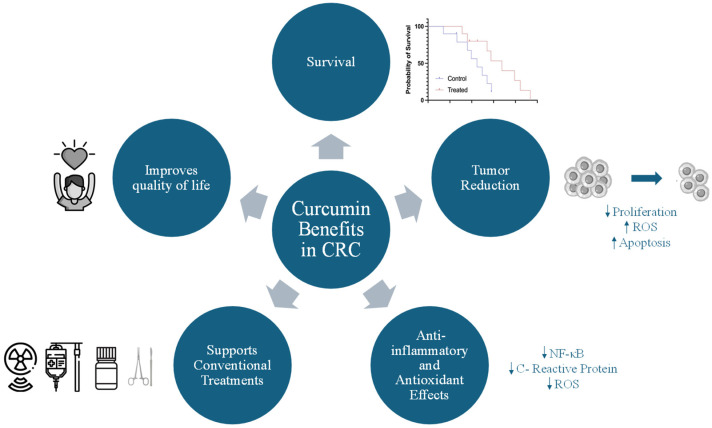
Benefits of curcumin-enriched diets in CRC treatment. ↑ Increase. ↓ Decrease.

**Figure 3 antioxidants-14-00388-f003:**
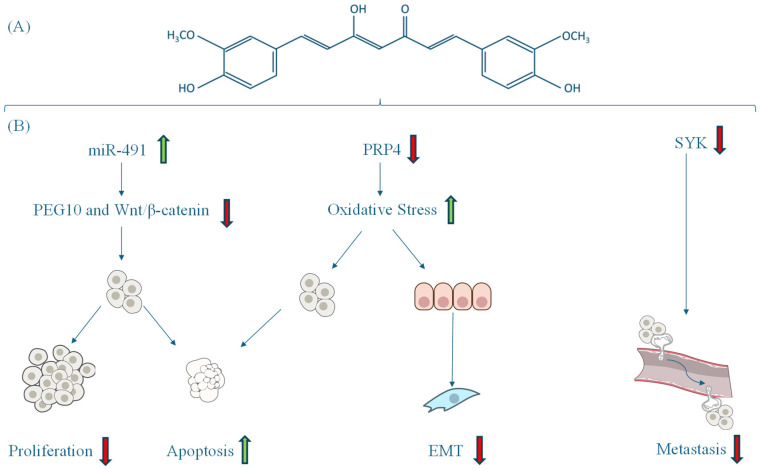
(**A**) Structural representation of curcumin. (**B**) Mechanisms regulated by curcumin in CRC: proliferation, apoptosis, oxidative stress, EMT, and metastasis. ↑ Increase. ↓ Decrease.

**Figure 4 antioxidants-14-00388-f004:**
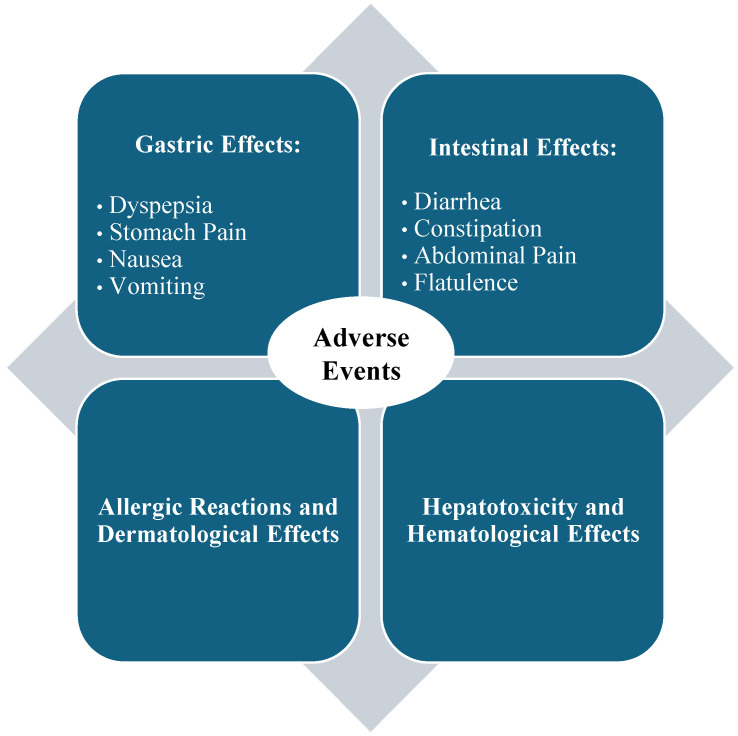
Adverse events of curcumin-containing products in CRC treatment.

**Table 1 antioxidants-14-00388-t001:** Characteristics of articles included.

First Autor	Year	Country Where the Study Was Conducted	Type of Study	Curcumin-Containing Product Administration	Chemotherapy/Radiotherapy	Results	Ref.
Clinical trials:							
Cruz-Correa	2018	Puerto Rico/USA	Double-blind, randomized controlled trial	3000 mg/day pure oral curcumin	NA	No significant reduction in polyp number or size compared to the placebo; curcumin was well-tolerated	[33]
Greil	2018	Austria	Phase I trial, dose-escalation	Intravenous liposomal curcumin was administered starting at a dose of 100 mg/m^2^ over 8 h, with subsequent escalation to 300 mg/m^2^ over 6 h	NA	Liposomal curcumin was well tolerated up to 300 mg/m^2^; no dose-limiting toxicity was observed. Anti-tumor activity was not detected, but significant tumor marker responses and clinical benefits were noted in two patients	[34]
Howells	2019	United Kingdom	Phase IIa randomized trial	Oral curcumin-containing product (Curcumin C3 Complex ^®^ (Sabinsa, East Windsor, NJ, USA) is 70–80% curcumin) (2 g/day)	FOLFOX/Bevacizumab	Improved overall survival with safe and tolerable combination therapy	[35]
Panahi	2021	Iran	Randomized double-blind placebo-controlled trial	500 mg/day orally for 8 weeks of curcumin-containing product (Curcumin C3 Complex ^®^ is 70–80% curcumin)	Chemotherapy	Significant reduction in erythrocyte sedimentation rate and C-reactive protein levels, slight decrease in IL-1α, and improvement in quality of life scores	[36]
Briata	2021	Italy	Randomized, double-blind, placebo-controlled, phase II presurgical trial	1 g daily curcumin-containing product (Meriva ^®^ (INDENA, Milan, Italy) is a complex of curcumin (20%) with soy phosphatidylcholine) combined with anthocyanins orally for 4–6 weeks	NA	Significant reduction in NF-κB and a trend toward Ki-67 reduction in adenoma tissues	[37]
Gunther	2022	USA	Phase II trial	4 g orally, twice daily of curcumin-containing product (Curcumin C3 Complex ^®^ is 70–80% curcumin) during chemoradiation therapy and for 6 weeks after	Capecitabine; radiation	Curcumin did not improve pathologic complete response rates. Bioavailability issues led to inconsistent plasma/tissue levels, limiting therapeutic efficacy	[38]
Nadeem	2023	India	Case-control study	Low and medium intake of mixed spices, including curcumin	NA	Curcumin, as part of mixed spices, may provide protective effects against gastrointestinal cancers at low–medium intake	[39]
Macis	2023	Italy	Presurgical trial	1 g of curcumin-containing product (Meriva^®^ is a complex of curcumin (20%) with soy phosphatidylcholine). Combined with anthocyanins orally	NA	The combined treatment of anthocyanins and curcumin did not directly alter circulating inflammatory and metabolic biomarkers. However, it exhibited a complex effect on biomarkers associated with colon cancer progression	[40]
Observational studies:							
Jeon	2022	South Korea	Prospective, observational, single-group analysis	Daily oral administration of 200 mg of curcumin in G-NLC	Bevacizumab/FOLFIRI	Improved median overall survival and progression-free survival; good tolerability and safety	[41]
Hoppe	2023	Germany	Cross-sectional observational study	Curcumin is one of the complementary and alternative medicine methods listed in the questionnaire distributed to participants	NA	The use of complementary and alternative medicine is linked to spiritual well-being and the need for meaning, highlighting the need for education to set realistic goals and avoid side effects. Diet shows positive psychological associations in cancer patients, though it is unclear whether diet influences psychological stability or vice versa	[15]
In vitro studies:							
Li	2018	China	In vitro	12.5 µM curcumin	NA	Curcumin upregulates miR-491, inhibits PEG10 and Wnt/β-catenin pathways, reduces proliferation, and promotes apoptosis in HCT-116 cells	[42]
Maletzki	2019	Germany	In vitro	20 µM curcumin for 72 h	Gemcitabine/Indoximod	Curcumin alone induced apoptosis and senescence in colorectal cancer cells. Combined with Gemcitabine, it enhanced tumor cell elimination significantly	[43]
Ahmed	2020	Republic of Korea	In vitro and in vivo study	30 µM curcumin (Sigma-Aldrich, St. Louis, MO, USA) for 24 h	NA	PRP4 overexpression conferred resistance to curcumin-induced apoptosis and promoted epithelial–mesenchymal transition (EMT). Deletion of PRP4’s kinase domain restored sensitivity to curcumin and inhibited EMT	[44]
Ex vivo study:							
Khan	2022	United Kingdom	Ex vivo explant model	Curcumin (Sigma, Gillingham, UK) concentrations 0–10 µM administered to tissue explants	NA	Curcumin induced apoptosis in tumor and stromal tissues, reduced immunosuppressive tumor microenvironment, increased T-cell movement toward cancer, and improved immune response	[45]
In silico study:							
Biswas	2021	Bangladesh	In silico study	Curcumin (molecular docking as inhibitor)	NA	Curcumin exhibited strong binding to SYK (spleen tyrosine kinase), suggesting its potential as a therapeutic candidate	[46]

NA = not applicable; G-NLC = ginsenoside-modified nanostructured lipid carrier; IL: interleukin; FOLFOX = combination chemotherapy regimen that used folinic acid, fluorouracil, and oxaliplatin; FOLFIRI = combination chemotherapy regimen that used folinic acid, fluorouracil, and irinotecan hydrochloride; EMT: epithelial–mesenchymal transition.

## Data Availability

The original contributions presented in this study are included in the article and Appendix A. Further inquiries can be directed to the corresponding authors.

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
