# Peer review of "The Potential Benefits of Curcumin-Enriched Diets for Adults with Colorectal Cancer: A Systematic Review"

_antioxidants, 2025, doi:10.3390/antiox14040388_

Round 1
Reviewer 1 Report
The topic is relevant in the field. Methodology of investigations seems to be correct. The conclusions are consistent with the evidence and arguments presented. The references are appropriate. While studying literature, I found a very similar publication titled “Anticancer Properties of Curcumin Against Colorectal Cancer: A Review”, doi: 10.3389/fonc.2022.881641, year: 2022. The publication submitted for evaluation can be treated as a supplement to the above, especially since only 3 publications are common. What is new is the limitation of studies only to adults, "On the other hand, the exclusion criteria for selecting research articles were: studies involving anticoagulant users, children, adolescents, or pregnant women; and studies that did not report original data". What is more, the conclusions in both publications are almost identical in terms of their meaning. Why was the search area limited to English and Spanish ? Nowadays, it is not a problem to translate any language into one that is understood? The authors write that "The systematic review was conducted between March and May 2024 using ..." That means that it was almost a year ago. In my opinion, the publication should be supplemented to the current state of knowledge. At the end of each paragraph, the authors briefly summarize it, presenting their opinion. However, it seems to me that such opinions are superficial and not sufficiently proven. I leave it to the editor to evaluate whether its novelty is sufficient to publish it.
Everything seems to be OK
Author Response
Antioxidants,
Editor and Reviewers,
We thank the editor and reviewers for revising our manuscript entitled “The Potential Benefits of Curcumin-Enriched Diets for Adults with Colorectal Cancer: A Systematic Review” by the authors María Jesus Neira, Constanza Valentina Mena, Keila Torres, and Layla Simón.
We are now submitting a modified document which includes the responses to the reviewers´ comments.
Reviewer 1:
-The topic is relevant in the field. Methodology of investigations seems to be correct. The conclusions are consistent with the evidence and arguments presented. The references are appropriate. While studying literature, I found a very similar publication titled “Anticancer Properties of Curcumin Against Colorectal Cancer: A Review”, doi: 10.3389/fonc.2022.881641, year: 2022. The publication submitted for evaluation can be treated as a supplement to the above, especially since only 3 publications are common. What is new is the limitation of studies only to adults, "On the other hand, the exclusion criteria for selecting research articles were: studies involving anticoagulant users, children, adolescents, or pregnant women; and studies that did not report original data". What is more, the conclusions in both publications are almost identical in terms of their meaning.
Response: Thank you for your comment. The literature review published by Ojo and cols. (2022) provides information about the use of curcumin in colorectal cancer, focused on its molecular mechanisms of action. However, our study presents a novel contribution by being the first systematic review to focus exclusively on adult populations, explicitly excluding studies involving children, adolescents, and anticoagulant users. Moreover, we provide updated data from 2018 to 2024 and discuss emerging formulations designed to enhance bioavailability. Additionally, our systematic review aims to answer the question: What are the beneficial effects of a curcumin-enriched diet in adult patients with colorectal cancer? In this sense, our focus was answering a clinical question and most of our outcomes were clinical. We also described the methods and the inclusion and exclusion criteria that prevent cherry-picking studies in a non-systematic way. These methodological rigor and transparency enhance the credibility, reproducibility, and reliability of our systematic review compared to traditional literature review.
-Why was the search area limited to English and Spanish? Nowadays, it is not a problem to translate any language into one that is understood?
Response: Thank you for your comment. We are English and Spanish speakers, so we preferred not to use other languages to avoid misunderstanding of the results and conclusions during translation.
-The authors write that "The systematic review was conducted between March and May 2024 using ..." That means that it was almost a year ago. In my opinion, the publication should be supplemented to the current state of knowledge.
Response: Thank you for your comment. We are sorry for the delay in sharing our manuscript, but we need some time to analyze and discuss the results. As you suggested, we added a recently published article in the new section “4.5. Current state of knowledge”. The systematic review of Gutsche et al. (2025) summarizes the current knowledge of curcumin in cancer treatment. However, Gutsche et al. included studies of head and neck, breast, prostate, and colorectal cancer, but their heterogeneity complicates their comparison and conclusion. Additionally, we have included research published in 2025 focus on CRC and curcumin.
-At the end of each paragraph, the authors briefly summarize it, presenting their opinion. However, it seems to me that such opinions are superficial and not sufficiently proven. I leave it to the editor to evaluate whether its novelty is sufficient to publish it.
Response: We thank you for your comments. We reformulate the conclusions within section 4.1.2 Tumor reduction, 4.1.4 Support to Conventional Treatment, 4.1.5 Quality of life, and 4.3. Adverse events of a curcumin diet in adult patients with CRC.
Reviewer 2 Report
The manuscript by Neira et al. is a literature review of the potential benefits of curcumin-enriched diets in adults with CRC. It is interesting work, but there is a major crux with the term ‘curcumin’ in the way it is used by the authors.
Curcumin is a chemical compound. The structure is shown in Figure 3A. However, the authors use ‘curcumin’ mainly for products, herbal supplements containing curcumin, that come from Curcuma rhizomes. Using the word ‘curcumin’ for such herbal products is confusing because they are not curcumin. They will contain a high percentage and will be standardized on a given content but they are not pure chemical substances.
The entire manuscript (including the abstract and conclusion) should, in my opinion, be adapted to this point. Clearly define the products used. For instance, Meriva® in Table 1 is an herbal supplement that contains more than curcuma alone. Also, in the discussion attention should be paid to clarity about the products used. Now there is a risk that outcomes of different products are compared that may probably not be compared because their composition differs.
Adapt lines 65ff. Curcumin is a main constituent of the rhizomes (not roots) of Curcuma longa L. (turmeric). Another important source of curcumin is Curcuma xanthorrhiza (or zanthorrhiza) Roxb. Both belong to the ginger family (Zingiberaceae). Note that Latin species names are printed in italic.
Line 69 contains a mistake. Alkaline conditions definitely do not stretch from pH 3 to 10!
Table 1, with the characteristics of the studies included, lists the studies chronologically starting with the most recent one. I would suggest grouping the studies according to the type of study (see lines 137ff), to prevent in vitro work is mixed with clinical studies. And order them chronologically per subgroup. The in vitro studies may have used pure curcumin, whereas patients will have used an herbal preparation.
Check the table on completeness and clarity regarding the products used. What, for instance, is G-NLC (Jeon 2022)? Pahini 2021: 500 mg/day of what?
Please explain what FOLFIRI and FOLFOX are.
Figure 4 shows drug interactions. This is not further discussed. Is this relevant here? If so, please discuss her (curcumin)-drug interactions and their possible clinical relevance.
Regarding the poor bioavailability of curcumin: is it possible that the effect in CRC comes from a local effect in the colon?
See above.
Author Response
Antioxidants,
Editor and Reviewers,
We thank the editor and reviewers for revising our manuscript entitled “The Potential Benefits of Curcumin-Enriched Diets for Adults with Colorectal Cancer: A Systematic Review” by the authors María Jesus Neira, Constanza Valentina Mena, Keila Torres, and Layla Simón.
We are now submitting a modified document which includes the responses to the reviewers´ comments.
Reviewer 2:
-The manuscript by Neira et al. is a literature review of the potential benefits of curcumin-enriched diets in adults with CRC. It is interesting work, but there is a major crux with the term ‘curcumin’ in the way it is used by the authors.
Curcumin is a chemical compound. The structure is shown in Figure 3A. However, the authors use ‘curcumin’ mainly for products, herbal supplements containing curcumin, that come from Curcuma rhizomes. Using the word ‘curcumin’ for such herbal products is confusing because they are not curcumin. They will contain a high percentage and will be standardized on a given content but they are not pure chemical substances.
The entire manuscript (including the abstract and conclusion) should, in my opinion, be adapted to this point. Clearly define the products used. For instance, Meriva® in Table 1 is an herbal supplement that contains more than curcuma alone. Also, in the discussion attention should be paid to clarity about the products used. Now there is a risk that outcomes of different products are compared that may probably not be compared because their composition differs.
Response: Thank you for your comments. As our systematic review aims to answer the question: What are the beneficial effects of a curcumin-enriched diet in adult patients with colorectal cancer?, our focus was to describe the clinical outcomes of a curcumin-containing product, independently of their source or composition. Following your suggestion, we have revised the entire manuscript to consistently use the terms “curcumin-containing product” or “curcumin-enriched diet” and have clarified their composition in Table 1.
-Adapt lines 65ff. Curcumin is a main constituent of the rhizomes (not roots) of Curcuma longa L. (turmeric). Another important source of curcumin is Curcuma xanthorrhiza (or zanthorrhiza) Roxb. Both belong to the ginger family (Zingiberaceae). Note that Latin species names are printed in italic.
Response: Thank you for your comment. We modified this sentence to include the missing source. Additionally, Latin species are now printed in italic.
-Line 69 contains a mistake. Alkaline conditions definitely do not stretch from pH 3 to 10!
Response: Thank you for your correction. We apologize for the mistake. Now, we explain better the pH conditions in which curcumin is degraded.
-Table 1, with the characteristics of the studies included, lists the studies chronologically starting with the most recent one. I would suggest grouping the studies according to the type of study (see lines 137ff), to prevent in vitro work is mixed with clinical studies. And order them chronologically per subgroup. The in vitro studies may have used pure curcumin, whereas patients will have used an herbal preparation.
Response: Thank you for your suggestion. We have re-ordered the studies according to their type and year of publication, starting with the most recent published. Additionally, we included the composition of the curcumin-containing products.
-Check the table on completeness and clarity regarding the products used. What, for instance, is G-NLC (Jeon 2022)? Pahini 2021: 500 mg/day of what?
Response: Thank you for your suggestion. We have now included the composition of each curcumin-containing product. Additionally, we have described which G-NLC means within the legend (Ginsenoside-modified Nanostructured Lipid Carrier).
-Please explain what FOLFIRI and FOLFOX are.
Response: Thank you for your suggestion. We included the meanings of FOLFIRI (combination chemotherapy regimen that used folinic acid, fluorouracil and irinotecan) and FOLFOX (combination chemotherapy regimen that used folinic acid, fluorouracil and oxaliplatin).
-Figure 4 shows drug interactions. This is not further discussed. Is this relevant here? If so, please discuss her (curcumin)-drug interactions and their possible clinical relevance.
Response: Thank you for your comment. “Drug interactions” referred to the combination of conventional chemotherapy with curcumin, but it does not settle with the other group of adverse events. So, we decided to delete this, but incorporate more information about gastric and intestinal effects, which are the most common.
-Regarding the poor bioavailability of curcumin: is it possible that the effect in CRC comes from a local effect in the colon?
Response: Thank you for your comment. We included this suggestion in the new section “4.5. Current state of knowledge”.
Round 2
Reviewer 1 Report
Citing the publication “Anticancer Properties of Curcumin Against Colorectal Cancer: A Review”, doi: 10.3389/fonc.2022.881641, year: 2022” and referring to it would allow for proving the novelty of the evaluated paper. This was not done, which is why I believe it has a low novelty.
No more.
Author Response
Antioxidants,
Editor and Reviewers,
We thank the editor and reviewers for revising our manuscript entitled “The Potential Benefits of Curcumin-Enriched Diets for Adults with Colorectal Cancer: A Systematic Review” by the authors María Jesus Neira, Constanza Valentina Mena, Keila Torres, and Layla Simón.
We are now submitting a modified document which includes the responses to the comments of reviewer 1.
Reviewer 1:
-Citing the publication “Anticancer Properties of Curcumin Against Colorectal Cancer: A Review”, doi: 10.3389/fonc.2022.881641, year: 2022” and referring to it would allow for proving the novelty of the evaluated paper. This was not done, which is why I believe it has a low novelty.
Response: Thank you for your comment. We included the literature review published by Ojo and cols. (2022) which provides information about the use of curcumin in colorectal cancer, focused on its molecular mechanisms of action that we added in the Introduction. Additionally, we explain how our systematic review tries to describe the clinical benefits of a curcumin-enriched diet in adult patients with colorectal cancer in a credible, transparent and reproducible manner.
We sincerely thank you for your invaluable suggestions that improve the quality of our manuscript. We hope to continue contributing with Antioxidants. Thank you once again for considering our work.
Regards,
Layla Simón Ph.D and Keila Torres Ph.D.
Corresponding authors
Reviewer 2 Report
The authors have modified the manuscript, considering my comments and suggestions. The current version is, in my opinion, acceptable for publication in Antioxidants.
No comments.
Author Response
Antioxidants,
Editor and Reviewers,
We thank the editor and reviewers for revising our manuscript entitled “The Potential Benefits of Curcumin-Enriched Diets for Adults with Colorectal Cancer: A Systematic Review” by the authors María Jesus Neira, Constanza Valentina Mena, Keila Torres, and Layla Simón.
We sincerely thank you for your invaluable suggestions that improve the quality of our manuscript. We hope to continue contributing with Antioxidants. Thank you once again for considering our work.
Regards,
Layla Simón Ph.D and Keila Torres Ph.D.
Corresponding authors